# Species-Specific Gamete Interaction during Sea Urchin Fertilization: Roles of the Egg Jelly and Vitelline Layer

**DOI:** 10.3390/cells11192984

**Published:** 2022-09-24

**Authors:** Nunzia Limatola, Jong Tai Chun, Luigia Santella

**Affiliations:** 1Department of Research Infrastructures for Marine Biological Resources, Stazione Zoologica Anton Dohrn, 80121 Napoli, Italy; 2Department of Biology and Evolution of Marine Organisms, Stazione Zoologica Anton Dohrn, 80121 Napoli, Italy

**Keywords:** fertilization, sea urchin eggs, polyspermy, actin, Ca^2+^ signaling, acrosome reaction, vitelline layer, species-specificity

## Abstract

In sea urchins, the sequence of the cellular and molecular events characterizing the fertilization process has been intensively studied. We have learned that to activate the egg, the fertilizing sperm must undergo morphological modifications (the acrosome reaction, AR) upon reaching the outer gelatinous layer enveloping the egg (egg jelly), which triggers the polymerization of F-actin on the sperm head to form the acrosomal process. The AR exposes bindin, an adhesive sperm protein essential for the species-specific interaction with the cognate receptor on the egg vitelline layer. To investigate the specific roles of the egg jelly and vitelline layer at fertilization of sea urchin eggs, *Paracentrotus lividus* eggs were incubated in acidic seawater, which removes the egg jelly, i.e., experimental conditions that should prevent the occurrence of the AR, and inseminated in the same medium. At variance with the prevailing view, our results have shown that these dejellied *P. lividus* eggs can still interact with sperm in acidic seawater, albeit with altered fertilization responses. In particular, the eggs deprived of the vitelline layer reacted with multiple sperm but with altered Ca^2+^ signals. The results have provided experimental evidence that the plasma membrane, and not the vitelline layer, is where the specific recognition between gametes occurs. The vitelline layer works in unfertilized eggs to prevent polyspermy.

## 1. Introduction

Sea urchin gametes have served as an exceptional animal model for over one hundred years in investigating the cellular and molecular aspects of fertilization and embryonic development. At variance with mammals, sea urchin eggs can be obtained in large numbers and used in the laboratory to study the physiological changes following the interaction of the fertilizing sperm with the egg surface in the experimental conditions that reproduce those occurring naturally at sea. Since the beginning of the investigation on the specific role of molecules (receptors) on the surfaces of sea urchin sperm and eggs, we have learned that to fertilize the eggs, sperm diluted in seawater must undergo profound morpho-functional changes upon reaching the extracellular gelatinous egg layer (egg jelly). The molecules of this layer induce exocytosis of the acrosomal vesicle of the sperm head and the formation of an F-actin process due to the polymerization of G-actin. The so-called ‘acrosome reaction (AR)’ requires external Ca^2+^ and an increase in the intracellular pH in sperm [1,2,3,4,5]. In the sea urchin, the sperm AR can also be induced by suspending the sperm in seawater at pH 9 without egg jelly and Na^+^ [6,7], or even by direct contact with solid surfaces [8]. However, even if the literature on the crucial role of the egg jelly in inducing the AR is pervasive (see the “fertilizin theory” on the agglutinating factor of the jelly layer material in solution introduced by Lillie, [9]), it has also been suggested that the egg jelly could exert an inhibitive effect on fertilization. Following the removal of the jelly layer in either mechanical or acidic conditions (seawater at a pH of 5.5–5.8), the dejellied *Paracentrotus lividus* eggs exhibited an improved fertilization rate when inseminated in normal seawater [10]. Furthermore, sea urchin sperm that were allowed to pass through the hull of the jelly coat isolated from the sea urchin eggs of different species showed no sign of the AR at the electron microscopy level [11]. Taking into account that the properties of the jelly layer components in solution may not necessarily be the same as in the jelly-intact eggs, the contradictory effects of the soluble egg jelly on the binding of sperm to acid-dejellied eggs of *Strongylocentrotus purpuratus* (showing a decrease) and *Lytechinus pictus* (showing an enhancement), were attributed to differences in the amount of jelly coat remaining on the surface of the eggs of these two species upon treatment with acidified seawater (at pH 4.7 for 3 min). It was demonstrated that the addition of soluble jelly to acid-dejellied *S. purpuratus* eggs, still possessing a sufficient amount of surface-associated jelly to promote the sperm AR, decreased the fertilizability of these eggs as a consequence of sperm isoagglutination. By contrast, the acid-dejellied eggs, having lost a considerable amount of surface-associated egg jelly, failed to induce the AR, which was restored by the addition of soluble jelly [12]. Recently, the role of the egg jelly in causing the AR in sea urchin sperm was confirmed by a model according to which sperm become activated upon their binding to sulfated polysaccharides of the jelly coat [13]. Following the AR in sea urchin sperm, bindin, an adhesive protein exposed on the acrosomal process, allows the species-specific binding of the fertilizing sperm with the egg receptors on the VL, and possibly a fusion with the egg plasma membrane [14,15]. Indeed, bindin isolated from the sperm acrosome was found to cause the aggregation of homospecific (and not heterospecific) sea urchin eggs. The characterization of the sperm binding receptor isolated from the VL of sea urchin eggs identified a conjugated protein with carbohydrate and lipid components [16]. In line with this, the treatment of unfertilized sea urchin eggs with enzymes or lectins that break down proteins or bind to the carbohydrate component of the egg glycoprotein sperm receptor prevented fertilization [17,18,19,20]. As for the molecular mechanisms mediating sperm–egg binding and fusion in sea urchins, it has been suggested that once exposed on the acrosomal process, bindin recognizes and interacts with the egg surface molecules identified as EBR1 (Egg Bindin Receptor 1) and a 350-KDa glycoprotein [16]. The acrosomal process perforates the VL and fuses with the plasma membrane. The finding that bindin KO sperm failed to bind to the VL covering the microvilli on the egg surface and activated the eggs of *Hemicentrous pulcherrimus* has added weight to the idea that bindin is essential for sea urchin fertilization [21].

It is noteworthy that different methods were devised to evaluate the role of the egg jelly in inducing the AR and to quantitatively analyze the sperm binding kinetics to dejellied sea urchin eggs in the absence or presence of soluble jelly. They included the cessation of the motility of the spermatozoa without injuring the eggs or a fixation of eggs at different time points after insemination [12,22,23,24]. However, the observation under a light microscope of the binding of the sperm with the egg surface or the elevation of the fertilization envelope (FE) as a sign of egg activation was is not sufficient to give accurate information on the ability of these bound sperm to transduce the signal as a result of their fusion with the egg plasma membrane.

Being intimately attached to the egg surface by various linkages, the VL has been thought to play a specific role in sperm–egg binding. Recent findings have shown that its integrity is essential to serve as a barrier to polyspermy in both homologous and heterologous fertilization [25]. A milder solubilization of the VL proteins by a dithiothreitol (DTT) treatment (10 mM in natural seawater at pH 7.65 instead of 9.2, the latter of which completely removed the VL [26]), also altered the Ca^2+^ signaling and F-actin remodeling in *P. lividus* eggs following fertilization, which led to impaired embryo development [25]. In this contribution, we analyzed the fertilization response of *P. lividus* eggs, whose egg jelly and VL had been removed, by following the previously shown methods. While the incubation of the sea urchin eggs in seawater at pH 5.5 for 5 min removed the egg jelly, the treatment of the eggs with DTT (10 mM in natural seawater at pH 9 for 20 min) solubilized the VL. [10,26]. At variance with the previous data in the literature in which the evaluation of the rate of fertilization of eggs deprived of their investments before insemination was determined by indirect methods (see above), the initiation of the sperm-induced Ca^2+^ signal was considered as the precise timing at which the sperm plasma membrane fused with that of the egg. The results have shown that unfertilized dejellied eggs inseminated in acidic seawater could still respond to the sperm but with altered Ca^2+^ responses, cortical granules exocytosis, and actin remodeling as a result of the effect of the acidic seawater–induced shrinkage, which injured the structural organization of the egg surface and cortex. The results have shown that many unreacted *P. lividus* sperm diluted in natural seawater (NSW) could bind and fuse with the plasma membrane of homologous eggs deprived of the VL, eliciting an altered Ca^2+^ response and polyspermic entry. By contrast, the heterologous sperm failed to fuse with the egg plasma membrane and transduce a proper Ca^2+^ signal. The results also indicate that at fertilization, the sea urchin sperm AR is not an absolute prerequisite for specific binding to the receptors on the VL and that a block to cross-fertilization resides at the level of receptors located on the egg plasma membrane.

## 2. Materials and Methods

### 2.1. Gametes Collection, Egg Jelly, and Vitelline Layer Removal and Fertilization Procedure

The *P. lividus* were collected from October to May in the Gulf of Naples and maintained at 16 °C in circulating seawater. The eggs were spawned by injecting 0.5 M of KCl into the body cavity. The released eggs were collected in natural seawater (NSW, pH 8.1) filtered with a Millipore membrane of 0.2 µm pore size (Nalgene vacuum filtration system, Rochester, NY, USA) and used for the experiments in the next 3 hr. The dry sperm were collected by pipetting from the male animal’s body and kept at 4 °C. A few minutes before insemination, the sperm were diluted in NSW at a final concentration of 1.84 × 10^6^ units/mL. In the experiments where dejellied eggs were used, the egg jelly was removed by placing the eggs in seawater titrated with HCl to pH 5.5 for 5 min. The eggs incubated in 1 mL of seawater at pH 5.5 or after washing with NSW were fertilized with 10 µL of sperm diluted in NSW. The removal of the VL was performed by incubating the eggs for 20 min in NSW containing a final concentration of 10 mM of DL-Dithiothreitol (DTT) (Sigma-Aldrich, St. Louis, MO, USA) at pH 9 adjusted with 1 mM of NaOH [26]. When needed, the eggs suspended in acidic seawater or after the removal of the VL were transferred to NSW for the fertilization experiments.

### 2.2. Scanning and Transmission Electron Microscopy (SEM & TEM)

For the SEM morphological analyses of the egg surface, unfertilized *P. lividus* eggs incubated in acidic seawater (pH 5.5) or suspended in seawater containing 10 mM of DTT (pH 9) to remove the VL [26], were fixed in NSW containing 0.5% glutaraldehyde (pH 8.1) for 1 h at room temperature. The samples were post-fixed with 1% osmium tetroxide in seawater for an additional hour and dehydrated in a series of increasing concentrations of ethanol. The subsequent critical point drying procedure was performed with a LEICA EM CP300. The specimens were then coated with a thin layer of gold using a LEICA EM ACE200 sputter coater and observed with a JEOL 6700F scanning electron microscope (Akishima, Tokyo, Japan). For the TEM observations, after the fixation of the eggs in NSW containing 0.5% glutaraldehyde (pH 8.1), the samples were post-fixed with 1% osmium tetroxide and 0.8% K_3_Fe(CN)_6_ for 1 h at 4 °C. After washing in NSW for 10 min twice, the samples were rinsed in distilled water for 10 min twice and finally treated with 0.15% tannic acid for 1 min at room temperature. After extensive rinsing in distilled water (3 times, 10 min each), the specimens were dehydrated in increasing ethanol concentrations. The residual ethanol was removed with propylene oxide before embedding in EPON 812. Ultrathin sections were cut with an ultramicrotome (Leica EM UC7) and observed under a Zeiss LEO 912 AB (Carl Zeiss Microscopy Deutschland GmbH) without staining.

### 2.3. Microinjection, Ca^2+^ Imaging, and Confocal Microscopy

Intact sea urchin eggs were microinjected using an air pressure transjector (Eppendorf Femto-Jet, Hamburg, Germany) as previously described [27]. To monitor the intracellular Ca^2+^ level changes following fertilization, 500 µM of Calcium Green 488 conjugated with 10 kDa of dextran were mixed with 35 µM of Rhodamine Red (Molecular Probes, Eugene, OR, USA) in the injection buffer (10 mM of Hepes, 0.1 M of potassium aspartate, pH 7.0) and microinjected into the eggs before insemination. The fluorescence images of cytosolic Ca^2+^ were captured with a cooled CCD camera (Micro-Max, Princeton Instruments) mounted on a Zeiss Axiovert 200 with a Plan-Neofluar 40/0.75 objective at about 3 s intervals, and the data were analyzed with MetaMorph (Universal Imaging Corporation, Molecular Devices, LLC, San Jose, CA, USA). Following the formula: F_rel_ = [F − F_0_]/F_0_, where F represents the average fluorescence level of the entire egg and F_0_ the baseline fluorescence, the overall Ca^2+^ signals were quantified for each moment and the F_rel_ was expressed as a RFU (relative fluorescence unit) for plotting the Ca^2+^ trajectories. Applying the formula: F_inst_ = [F_t_ − F_(t − 1)_]/F_(t − 1)_, the instantaneous increment of the Ca^2+^ level was analyzed to locate the specific area of momentary Ca^2+^ increase. The values of the Ca^2+^ signals were obtained from three independent experiments (N), and the number of eggs (n) being analyzed for each condition is specified in the Results section. To visualize the F-actin in living eggs, 10 µM (pipette concentration in methanol) of AlexaFluor 568-phalloidin (Molecular Probes) or 10 mg/mL of bacterially expressed LifeAct-GFP fusion protein were microinjected into the unfertilized eggs in two independent experiments (N). The eggs microinjected with the fluorescent actin probes were observed with a Leica TCS SP8X confocal laser scanning microscope equipped with a white light laser and hybrid detectors using the Lightning deconvolution mode (Leica Microsystem, Wetzlar, Germany). The number of eggs examined for each condition (n) is specified in the Results section.

### 2.4. Visualization of Sperm inside the Eggs upon Insemination

The diluted sperm were stained with 5 µM of Hoechst-33342 (Sigma–Aldrich, Saint Louis, MO, USA) for 30 s before insemination. The labeled decondensed sperm nuclei incorporated into the *P. lividus* eggs were counted after their visualization in the cytoplasm of fertilized eggs 5 min after insemination using a cooled CCD (Charge-Coupled Device) camera (MicroMax, Princeton Instruments Inc., Trenton, NJ, USA) mounted on a Zeiss Axiovert 200 microscope with a Plan-Neofluar 40X/0.75 objective and with a UV laser. The number of fertilized eggs examined (n) for each condition in two independent experiments (N) is shown in the Results section.

### 2.5. Statistical Analysis

The numerical MetaMorph data were compiled and analyzed with Excel (Microsoft Office 2010) and reported as mean ± standard deviation (SD) in all cases in this manuscript. A one-way ANOVA was performed through Prism 5 (GraphPad Software), and *p* < 0.05 was considered statistically significant. For the ANOVA results showing *p* < 0.05, the statistical significance of the difference between the two groups was assessed by Tukey’s post hoc tests. The two groups of data showing significant differences from each other were marked with symbols indicating the *p* values in the figure legends.

## 3. Results

### 3.1. The Removal of the Egg Jelly and Vitelline Layer of P. lividus Eggs Has a Striking Effect on the Egg Surface Morphology

The scanning and transmission electron (SEM and TEM) micrographs of Figure 1 depict a lower and higher magnification of the outer surface of unfertilized *P. lividus* eggs before and after the removal of the jelly and the vitelline layer. Note that the transparent egg jelly which normally surrounds the eggs was not preserved during the fixation procedure for the SEM observations. In the unfertilized control eggs (Figure 1A-C), the vitelline layer (VL) was intimately associated with microvilli (MV) protruding from the egg surface (Figure 1B), as shown in the TEM micrograph in panel C. Figure 1C also shows the cortical granules (CG) beneath the egg plasma membrane. The suspension of intact unfertilized eggs in seawater at pH 5.5 (acidic seawater) for 5 min not only removed the egg jelly but also had a striking effect on the morphology of the egg surface. At variance with the control eggs suspended in normal seawater at pH 8.1, the acidic seawater incubation induced the shrinkage of the egg surface (Figure 1D), which was also visible at a higher magnification (Figure 1E). The VL of the eggs incubated in acidic seawater still covered the microvilli on the egg surface, as shown in the TEM micrograph (Figure 1F). The shrinkage effect on the egg surface induced by 5 min of the acidic seawater treatment was reversed when the eggs were transferred to NSW at pH 8.1 (data not shown). Our previous studies showed that pretreatment of *P. lividus* eggs with 10 mM of DTT in NSW (pH 7.57) induced changes in the microvillar morphology and an alteration of the VL structure without removing it [25]. To completely solubilize the VL, the *P. lividus* eggs were incubated in NSW containing 10 mM of DTT (pH 9) for 20 min. The SEM micrograph in Figure 1G shows that, while the DTT treatment induced no visible change in the cell volume, a dramatic alteration of the microvillar morphology and egg surface was evident at a higher magnification (Figure 1H). The structural modification of the microvilli induced by the incubation of the eggs in the DTT also included its retraction in some regions of the egg surface. The complete removal of the VL is well evident in the TEM image (Figure 1I), where the VL does not cover the plasma membrane. The TEM image shows cortical granules in the egg cortex beneath the plasma membrane.

### 3.2. Alteration of the Cortical Reaction upon Insemination of P. lividus Eggs Deprived of the Jelly and Vitelline Layers

Figure 2A,B shows SEM micrographs at lower and higher magnifications of the structure of the fertilization envelope (FE) surrounding the surface of *P. lividus* eggs fertilized in NSW (control). Five minutes after insemination, the TEM image in Figure 2C shows the formation of the hyaline layer (HL) and the thickening of the FE following the release of the content of the CG into the perivitelline space (PS). A clear alteration of the cortical reaction of *P. lividus* eggs incubated for 5 min in acidic seawater (pH 5.5) and fertilized in the same medium is shown in Figure 2D–F. At variance with the control in Figure 2A, the SEM micrograph of Figure 2D shows that, 20 min after insemination, numerous sperm were still bound to the structurally altered FE as compared to that in the control (Figure 2A,B), probably as a result of the retraction of the microvilli for the shrinkage of the egg induced by the incubation in acidic seawater prior to insemination (Figure 1D). The TEM micrograph (Figure 2F) also shows that even if the CG underwent exocytosis as they were no longer seen beneath the plasma membrane, the FE failed to undergo a full elevation from the egg surface. The attachment of some parts of the discontinuous and thinner FE with the elongated microvilli in the perivitelline space prevented its separation from the egg surface, as shown in the lower magnification image in Figure 2F.

The SEM and TEM micrographs in Figure 2G–I show the surface of a *P. lividus* egg fertilized in NSW after 20 min of treatment with DTT in NSW at pH 9 to remove the VL and the jelly coat to which the latter is attached. The low magnification in Figure 2G shows that the removal of both layers from the egg plasma membrane favored polyspermic fertilization. Five minutes after the insemination of denuded eggs, two sperm were seen in the process of being incorporated into the egg by the long microvilli of the fertilization cones. Five minutes after insemination, elongated microvilli, which are normally formed due to the cortical F-actin polymerization, were easily visible both in the SEM (Figure 2H) and TEM images (Figure 2I) on the activated egg surface due to the lack of formation and elevation of the FE.

### 3.3. P. lividus Eggs Fertilized in Seawater at pH 5.5 or in NSW after DTT Treatment at pH 9 Show an Altered Sperm-Egg Binding and Entry into the Eggs

The effect of the altered egg surface and microvillar morphology on the molecular mechanisms of sperm–egg binding was investigated by looking at the sperm entry (5 min after insemination) into the eggs deprived of egg investments. Figure 3A shows fluorescent and transmitted light images of monospermic fertilization experienced by all the control eggs (n = 40, inseminated in NSW at pH 8.1). All the fertilized eggs underwent CG exocytosis and elevation of the FE. At variance with this, only 14 out of 40 eggs incubated in seawater at pH 5.5 to remove the egg jelly and inseminated in the same medium showed monospermic fertilization. The lower number of sperm incorporated into the eggs fertilized in acidic seawater could be linked to the failure of detecting sperm inside the eggs five minutes after insemination, considering the delay employed by the sperm to bind, fuse and transduce the Ca^2+^ signals in these experimental conditions (Figure 4). Note that at variance with the transmitted light image of the control in Figure 3A, even if the FE was formed when the eggs were inseminated in acidic seawater (Figure 3B), it did not fully elevate from the egg surface and it was, therefore, possible to visualize it only at the electron microscopy level (Figure 2F).

By contrast, the denuded eggs, after being treated with DTT for 20 min at pH 9, washed several times in NSW, and then inseminated in NSW at pH 8.1, were penetrated by multiple sperm (34 out of 40 eggs, with an average of 3.3 ± 1.7 sperm per egg), (Figure 3C) possibly as a result of the alteration of the microvillar and egg surface topography (Figure 1H and the graphical abstract).

### 3.4. The Removal of the Vitelline Layer Strongly Alters the Ca^2+^ Response at Fertilization

Previous studies from our laboratory have amply shown that the sperm-induced Ca^2+^ response in sea urchin eggs is strongly linked with the microvilli morphology and structural organization of the egg cortex. In normal conditions, upon insemination, the eggs experience the first Ca^2+^ signal, the so-called cortical flash (CF), which takes place simultaneously at the periphery of the egg cortex as a result of the activation of the L-type Ca^2+^ channels promoting a Ca^2+^ influx. The CF is followed by a Ca^2+^ wave occurring during the separation of the VL from the egg surface due to the CG’s exocytosis [28]. The relative fluorescence images of Figure 4A show the Ca^2+^ signals at fertilization in NSW induced by two sperm in an egg deprived of the VL (and egg jelly), which in this case were elicited on the opposite sides of the egg. The graphs of Figure 4B show the fertilization Ca^2+^ increased in the control eggs fertilized in NSW at pH 8.1 (green color), eggs fertilized while incubated in acidic seawater (blue color), and eggs inseminated in NSW at pH 8.1 following the treatment with DTT for 20 min at pH 9 (red color). Both the graphs and histograms show striking differences in the pattern of the sperm-induced Ca^2+^ signals in the eggs inseminated in acidic seawater as judged by the remarkably delayed time required by the sperm until they initiated the CF (arrows in the graphs) as a sign of its fusion with the egg plasma membrane (547 ± 217.5 s, n = 22), which was nearly 10 times longer than that in the control eggs fertilized in NSW at pH 8.1 (54 ± 30 s, n = 38, and *p* < 0.01). When fertilized in acidic seawater, a complete inhibition of the Ca^2+^ response was observed in 6 out of 28 eggs; thus, in general, the sperm had difficulties initiating Ca^2+^ responses in the eggs incubated in acidic seawater. Even when the CF was generated, its amplitude was significantly lower (0.053 ± 0018 RFU, n = 22) than that of the control eggs (0.084 ± 0.017 RFU, n = 38, and *p* < 0.01) (Figure 4C). Similarly, the average peak amplitude of the Ca^2+^ wave in all the eggs incubated and fertilized in acidic seawater was significantly lower (0.43 ± 0.03 RFU, n = 22) than that of the control (0.65 ± 0.07 RFU, n = 38, and *p* < 0.01).

When the denuded eggs, i.e., deprived of both the VL and jelly as a result of the DTT treatment at pH 9, were inseminated after being washed several times in NSW at pH 8.1, they all responded to the sperm with altered Ca^2+^ responses. The graphs in Figure 4B (red color) show that the sperm-induced Ca^2+^ increases in the stripped eggs were significantly smaller than in the control (green color). In contrast, no significant difference was detected in the time lag between the insemination and the initiation of the CF. Nevertheless, 6 out of 25 denuded eggs did not experience any CF upon insemination, possibly due to the compromised functionality of the Ca^2+^ influx channels as a result of the morphological changes of the microvilli where the channels reside [28]. When multiple sperm elicited the Ca^2+^ responses in the egg, the heights of both the CF and Ca^2+^ wave were highly reduced in comparison with the control eggs (Figure 4C). It is noted that the Ca^2+^ wave propagated more slowly to reach the antipode in the denuded eggs (79 ± 21 s versus 35 ± 9.5 s of the control, with a *p* < 0.01).

### 3.5. The Plasma Membrane of Denuded Eggs of P. lividus Does Not Fuse with Arbacia lixula Sperm

It is widely accepted that sea urchin sperm must first undergo a species-specific AR after reaching an egg’s jelly for successful fertilization. This reaction is a prerequisite for sperm–egg binding and fusion with the egg plasma membrane. To determine the precise stage of fertilization at which gamete incompatibility is expressed during cross-fertilization, whole and denuded *P. lividus* eggs were inseminated in NSW with *A. lixula* sperm whose DNA was stained with Hoechst 33342 to eventually visualize them inside the eggs. The capability of the heterologous sperm of *A. lixula* to transduce its Ca^2+^ signals was also tested. The fluorescent and transmitted light images of Figure 5A show that when intact *P. lividus* eggs were cross-inseminated with labeled *A. lixula* sperm, the sperm were seen to be embedded in the jelly layer surrounding the eggs and around the egg surfaces. The graph of Figure 5A shows the Ca^2+^ signals that were induced following the insemination of whole *P. lividus* eggs with *A. lixula* sperm. An analysis of the Ca^2+^ changes in the egg as a result of the fusion between the plasma membranes of the two gametes showed that out of 17 eggs, only three responded to the sperm with a CF of a smaller amplitude as compared to that elicited by homologous sperm (Figure 4B green lines), which was not followed by a Ca^2+^ wave, indicating only a transient continuity between the plasma membrane of the two gametes.

At variance with the intact eggs, when denuded *P. lividus* eggs were inseminated in NSW with fluorescent *A. lixula* sperm, no binding was detected on the egg plasma membrane (Figure 5B). A comparison of the sperm-induced Ca^2+^ signal in heterospecific crosses of *A. lixula* sperm with whole (see the graph in Figure 5A) and denuded eggs of *P. lividus* shows that during 20 min of measurement, the numerous (n = 63 with an average of 7.5 ± 3.3 CF per egg) and smaller CF elicited in 18 out of 26 denuded eggs upon the addition of *A. lixula* sperm (see the graph in Figure 5B), can be interpreted as Ca^2+^ releasing events in the egg surface induced by the mechanical stimulation of the swimming sperm. SEM observations of the denuded *P. lividus* fixed at different times after heterospecific insemination confirmed the absence of sperm bound to the egg surface (data not shown).

### 3.6. Alteration of the Cortical Actin Polymerization and Translocation following Incubation and Insemination of P. lividus Eggs in Acidic Seawater

A wave of cortical actin polymerization following fertilization and F-actin translocation from the fertilized egg surface to the center have previously been detected in the living eggs of different sea urchin species using fluorescent actin probes [29,30]. In a series of previous studies, we demonstrated that even if the alteration of the cortical actin of unfertilized *P. lividus* eggs by several agents compromised the translocation of actin fibers following the fertilization of *P. lividus* eggs [25,31,32,33,34], the usage of fluorescent phalloidin did not allow the detection of the actin polymerization in the egg cortex following insemination. To this aim, a LifeAct-GFP fusion protein was microinjected into unfertilized *P. lividus* eggs [35] to follow the polymerization of actin in the cortical region induced by a sperm stimulation in a control and in eggs incubated and inseminated in acidic seawater. Figure 6 shows a confocal time-lapse acquisition (20 min) of the actin changes in the egg cortex following the insemination of *P. lividus* eggs in normal and acidic seawater experimental conditions. Five minutes after the sperm addition, a dense actin layer was visualized in the cortical region. A spike formation in the perivitelline space (PS) accompanying the elevation of the FE was also visible 20 min after insemination (n = 13) in the overlay image. Even if it was still possible to visualize a layer of actin in the eggs incubated and fertilized in acidic seawater, an evident inhibition of the polymerization of actin was shown in the confocal image captured 5 min after insemination. Actin polymerization, albeit much thinner than in the control eggs, was manifested 20 min after the insemination of eggs in acidic seawater, showing an impaired elevation in the FE (n = 6). Another sign of the alteration of the egg surface by the acidic treatment was the presence of sperm attached to the “sticky” egg surface 20 min after insemination, usually not seen in the control eggs at fertilization (see also Figure 2D). The visualization of the F-actin translocation using AlexaFluor 568-phalloidin microinjected in unfertilized *P. lividus* eggs confirmed the mobilization of actin fibers as a consequence of a normal cortical actin remodeling, as shown in the confocal images of Figure 7 (n = 10). Interestingly, the mobilization of the actin filaments towards the center of the zygote, which was evident 20 min after insemination, coincided with the polymerization of the non-filamentous cortical actin layer, detectable only with the LifeAct-GFP fusion protein (Figure 6). Finally, in the eggs treated and fertilized in acidic seawater, the centripetal migration of the actin fibers was impaired (Figure 7, n = 6).

## 4. Discussion

The initial act in the chain of reactions during the fertilization process is the interaction between the fertilizing sperm and the egg surface. This ultimately leads to the union of the chromosomes carried by the sperm and the egg pronucleus. Information on the molecular mechanism regulating the sperm–egg interaction has been mainly based on research on marine animal models, such as sea urchins and starfish, that reproduce by external fertilization. To ensure monospermic fertilization, sea urchin eggs seem to operate both a fast electrical block to polyspermy, which starts as soon as the sperm fuses with the plasma membrane [36] and a mechanical block provided by the separation of the vitelline layer from the egg plasma membrane following the dehiscence of the cortical granules [37]. The release of a trypsin-like protease not only promotes the detachment of sperm from the VL but also prevents the binding of additional ones by destroying the sperm receptors [38]. However, recently it has been shown in *P. lividus* eggs that the fast block to polyspermy is not electrically but structurally mediated [31], as initially suggested more than one hundred years ago [39,40]. In addition to the opportunity to study the structural, electrical, and Ca^2+^ changes occurring on the egg surface upon insemination in seawater [39,41,42,43], the large availability of eggs in these two species has allowed biochemical studies to characterize the molecules (receptors) being involved in the species-specific binding of the sperm with the egg surface that is critical to preserving a species identity. Among the different strategies adopted by marine organisms to prevent interspecific fertilization, the AR represents a widespread phenomenon occurring not only in invertebrates but also in primitive and higher vertebrates; thus, interesting from the phylogenetic point of view [44]. In this regard, starfish (a more ancient group that is separated from sea urchins by about 500 million years) represent the system *par excellence* where the sequence of events leading to the egg jelly-dependent formation and function of the acrosomal filament (20 µm in length) has been well characterized in vitro and in vivo [8,45,46]. The fusion of the tip of the acrosomal filament with the egg plasma membrane was shown by the electrical changes of the egg plasma membrane at fertilization [47], and by the visualization of the Ca^2+^ increases in the egg cortex when the head of the fertilizing sperm was outside the egg jelly [5,48,49,50]. At variance with starfish, in homologous sea urchin fertilization, the site of the formation of the acrosomal filament and the process of its fusion with the egg plasma membrane, ensuring the response of the egg cortex, have been primarily examined in fixed samples at the light and electron microscope levels due to the abridged length of the acrosomal process [3,51,52]. Furthermore, electrophysiological measurements aimed at examining the attachment and fusion of the acrosome-reacted sperm with the eggs of another species were carried out on sea urchin eggs that had been allowed to adhere to plastic Petri dishes pretreated with protamine sulfate and inseminated often with a voltage clamp [53]. In this experimental condition, the egg surface was inevitably altered, which might have compromised the mode of the egg’s interaction with the fertilizing sperm.

A critical comparative analysis of data from the literature on the adaptive variety of the molecular mechanisms regulating the sperm–egg interaction in the course of evolution led to the suggestion that, as with mammalian and ascidians’ eggs, sea urchin eggs also have the VL and not the egg jelly as the site at which the fertilizing sperm undergoes the AR [54,55,56]. Thus, the VL and the AR should have undergone a parallel evolution because the fertilizing sperm lacking an acrosome can reach and fuse with the egg plasma membrane through the micropyle opening of the thick vitelline coat (chorion) in otheranimal species [57,58]. However, it must be said that it has been recently shown that the AR in mouse sperm is initiated before contacting the zona pellucida (ZP), which is equivalent to the VL; the acrosome-reacted sperm are able to pass through both the cumulus and ZP of other eggs [59]. The suggestion that, also in the sea urchin, the sperm AR should result from the interaction between the receptors on the egg VL and those on the sperm plasma membrane is derived from the studies on the species-specificity of sperm binding and fertilization in heterospecific insemination [60,61]. At variance with this view, a comparison among the onset of the initial Ca^2+^ signal following sperm–egg membranes’ fusion in control and dejellied eggs suggested that neither the egg jelly nor the VL is the site at which *P. lividus* sperm undergo the AR. Indeed, the results of this contribution have shown that the sperm are still able to fertilize the eggs incubated in acidic seawater, which should prevent the polymerization of the actin of the acrosomal process, known to be induced by a pH increase. A considerable time delay between the addition of the sperm and the onset of the first Ca^2+^ is attributable to microvillar morphology changes and shrinkage of the egg, causing an alteration of the sperm-induced Ca^2+^ signals (Figure 4B blue lines). These results are in line with previous findings highlighting the crucial role of microvilli and cortical granules structures in regulating the Ca^2+^ response at fertilization of this sea urchin species [25,27,28,31,32,33]. The lack of actin polymerization is another indication of the injury of the egg surface induced by acidic seawater together with the inhibition of the translocation of actin fibers from the zygote surface to the center of the egg (Figure 6 and Figure 7). Interestingly, the effect of acidic seawater in lowering the actin polymerization following fertilization counteracts the induction of microvilli elongation following the actin polymerization promoted by incubating the unfertilized *P. lividus* eggs in seawater titrated to pH 9 by NH_4_OH [34]. These findings suggest that intracellular pH changes regulate the morpho-functional aspect of the structural dynamics of the cortical actin, as corroborated by the results showing the recovery of the shrinkage and surface topography modification of eggs treated with acidic seawater upon their transfer back to seawater at a pH of 8.1. Such morphologically-recovered eggs also experienced a normal sperm-induced Ca^2+^ response upon insemination (data not shown).

With regard to the fertilization response still occurring in the eggs incubated in the acidic seawater, the results above may indicate that remnants of jelly around the egg or in the seawater could still induce the sperm AR, independently of the lower pH of the seawater. However, the results of this contribution have shown that eggs devoid of the VL following its solubilization by a DTT treatment at pH 9, which also removes the egg jelly attached to it, can be instead fertilized by multiple sperm after a series of washing with NSW, ensuring that there are no traces of both layers. The insemination of denuded eggs with unreacted sperm produces a fertilization response with modes that strictly reflect the dramatic effect of the removal of the VL on the structural organization of the microvilli and regions of the egg plasma membrane (see the graphical abstract). The morphological modifications of the egg surface promote the fusion of multiple sperm, generating impaired Ca^2+^ signals (Figure 4B red lines). Unfortunately, results on the alteration of the F-actin remodeling following the fertilization of denuded eggs are missing due to the difficulty of keeping eggs injected with AlexaFluor 568-Phalloidin or LifeAct-GFP fusion protein alive after a treatment with DTT (pH 9) to monitor the cortical actin changes following fertilization; however, results on the effect of a less basic DTT treatment of the eggs (pH 7.57 instead of 9) on the F-actin reorganization following fertilization have already been published [25]. Interestingly, when denuded *P. lividus* eggs were cross-fertilized with *A. lixula*, there was no sign of a direct plasma membrane fusion between the gametes of the two species (Figure 5B). Thus, at variance with the data in the literature, the results from this work suggest that the intact vitelline layer covering the egg plasma membrane of unfertilized eggs acts as a barrier for homologous and heterologous fertilization, as it prevents the fusion of additional sperm with the egg plasma membrane. Then, how can an unreacted sperm fuse with the egg plasma membrane and transduce the fertilization signals? It is well known that the mature eggs of sea urchin collected from the same animal exhibit different fertilizability due to the variability of the precise timing of the polar bodies extrusion, the hallmark of the two meiotic divisions. It must, therefore, be possible that sperm could also be endowed with morpho-functional differences, making them unequally capable of fertilizing the eggs. In this regard, it has been reported that 3 to 8% of the sperm population already possess a short acrosomal filament when diluted in seawater without egg jelly [7,62].

Our observations may suggest that, at fertilization, the binding and fusion of the *P. lividus* sperm to the egg plasma membrane occurs at a preferential site uncovered by the vitelline layer. An explanation that sounds plausible for the recent finding in starfish is that the extension of the long acrosomal filament is formed to span the thickness of the egg jelly to reach and fuse with the egg plasma membrane through pre-existing openings on the VL [48,49,50]. The results of this contribution suggest the presence of specialized fusogenic sites in the unreacted sea urchin sperm plasma membrane too, that recognize those on the egg plasma membrane, as is seen in the mating of ancestral unicellular cells and organisms [55]. It follows that the differentiation of the VL around the eggs in the course of evolution had the function of masking the multiple fusogenic sites of the egg plasma membrane to prevent polyspermy which would lead to an abnormal development. Indeed, polyspermic fertilization in *P. lividus* is achieved when the structural integrity of the VL of the unfertilized eggs is altered [25,34] or completely removed, as shown in this contribution.

## 5. Conclusions

Despite the significant structural diversity of eggs and sperm of the animal species and the different physiological behaviors, the fundamental event of the fertilization process, which depends on species-specific recognition, is the fusion of the sperm with the egg plasma membrane to transduce the fertilization signals. At the fertilization of sea urchin eggs, the species-specific molecular events regulating the gamete interactions have highlighted the crucial role of the sperm receptors on the vitelline layer for binding the acrosome-reacted sperm exposing the adhesive protein, bindin, for fusion with the egg plasma membrane. At variance with the prevailing view in the literature, the results of this contribution have shown that restricted regions of the egg plasma membrane are differentiated to fuse with the plasma membrane of acrosome-unreacted homologous, and not heterologous, sperm. Thus, the integrity of the structure of the vitelline layer covering the surface of unfertilized eggs is essential to mask the fusogenic sites residing in the plasma membrane, the exposure of which would induce polyspermic homologous fertilization. Our results on the requirement of specific receptors on the plasma membrane, and not the vitelline layer, for the specific recognition and fusion with sperm may help us to understand the role played by the vitelline layer in the eggs of different species to seek common basic molecular mechanisms regulating fertilization in other animal groups.

## Figures and Tables

**Figure 1 cells-11-02984-f001:**
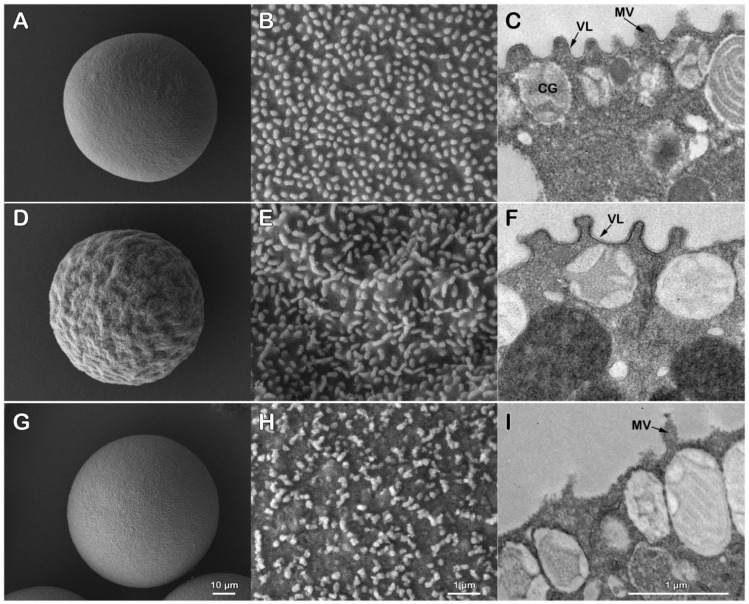
Alteration of the egg surface and microvillar morphology following incubation of *P. lividus* eggs in acidic seawater (pH 5.5) and after DTT treatment (pH 9) to remove the vitelline and egg jelly. SEM micrographs at lower (**A**) and higher magnification (**B**) of the outer surface of *P. lividus* eggs fixed after incubation in NSW (pH 8.1). Note the microvilli are regularly arranged on the smooth outer surface of the eggs. TEM observations (**C**) reveal the vitelline layer (VL) covering the microvilli (MV) in the absence of the jelly coat, which was not preserved in the eggs following the fixation procedure. Cortical granules (CG) are seen attached to the egg plasma membrane. Five minutes of incubation of the eggs in acidic seawater induced morphological alteration of the egg cortex, evidenced by the shrinkage of the cell volume shown in the SEM images in (**D**,**E**). The ultrastructural analysis revealed that, after incubating the eggs in acidic seawater, they still possessed the VL attached to the egg plasma membrane (**F**). The incubation of the eggs in NSW containing 10 mM of DTT for 20 min (pH 9) induced a dramatic alteration of the surface of the egg and microvillar morphology. SEM micrographs in (**G**,**H**) show that such treatment heavily damaged the egg surface and reduced the number of microvilli without changing the cell volume. In addition to the structural alteration of the microvilli, the DTT treatment completely removed the VL from the egg plasma membrane, as shown in the TEM image (**I**).

**Figure 2 cells-11-02984-f002:**
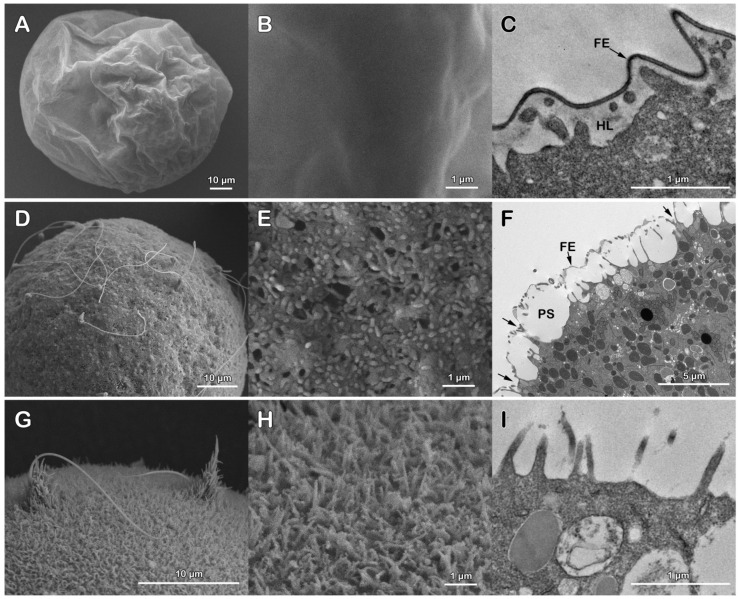
Effect of acidic seawater and DTT treatment to remove the vitelline layer on the cortical reaction of *P. lividus* eggs at fertilization. The SEM micrographs (**A**,**B**) show at a lower and higher magnification the fertilization envelope (FE) 5 min after insemination of a control egg in NSW (pH 8.1). At this time of the fertilization process, the continuous and thick FE, as a result of the exocytosis of the cortical granules (CG) in the perivitelline space (PS), is well evident in the TEM image in (**C**). Long microvilli embedded in the hyaline layer (HL) are visible. An alteration of the cortical reaction is evident in both SEM (**D**,**E**) and TEM micrographs (F), showing the surface of a *P. lividus* egg incubated for 5 min in acidic seawater and inseminated in the same medium. The eggs were fixed 20 min after sperm addition. Note in (**D**) that at this time of the fertilization process, the sperm were still bound to the FE, whose structure was heavily altered by the shrinking effect of the acidic seawater on the structural organization of the egg surface and cortex. At variance with the control, the thinner FE surrounding the fertilized egg in the TEM image in (**F**) was ruptured and failed to fully elevate from the egg surface due to its attachment with the elongated microvilli (arrows) in the perivitelline space (PS). The SEM micrograph in (**G**) shows the long microvilli of the fertilization cones engulfing two sperm on the surface of an egg deprived of both the vitelline layer and egg jelly by the DTT treatment prior to insemination. Note that the absence of the elevation of the FE in the SEM (**H**) and TEM (**I**) images allowed the visualization of the elongation of microvilli (MV) which usually occurs in the perivitelline space. See Figure 2A–C for comparison with the control.

**Figure 3 cells-11-02984-f003:**
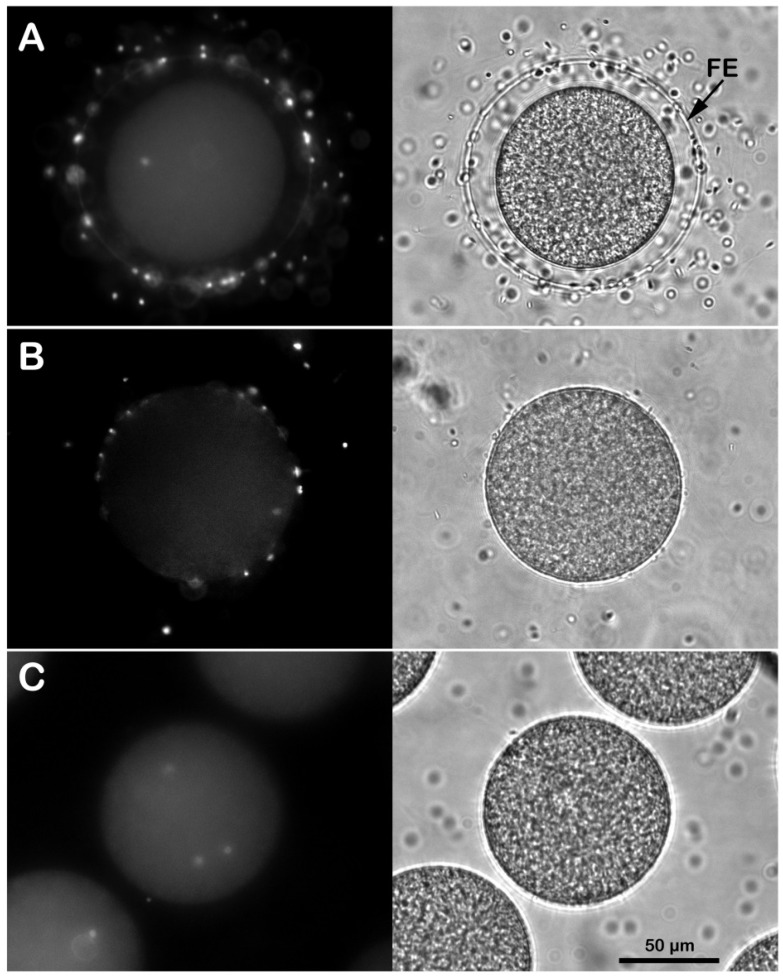
Effect of the removal of the egg jelly and vitelline layers on the sperm entry. (**A**) Intact *P. lividus* eggs inseminated in NSW with DNA-stained sperm were all penetrated by a single one, as shown by the UV laser in epifluorescence microscopy 5 min after insemination. The FE was fully elevated around the zygote, as shown in the transmitted light image. Monospermic fertilization or inhibition of sperm entry was observed in *P. lividus* eggs incubated for 5 min in acidic sea water and fertilized in the same medium (**B**). Note that at variance with the control in A, it was difficult to visualize the elevation of the FE in the eggs fertilized in acidic seawater due to its attachment to some regions of the egg surface, as shown in the EM image in Figure 2F. Multiple sperm incorporation was observed when the *P. lividus* eggs were pre-treated with 10 mM of DTT at pH 9 for 20 min to remove the egg’s layers and then fertilized in NSW (**C**).

**Figure 4 cells-11-02984-f004:**
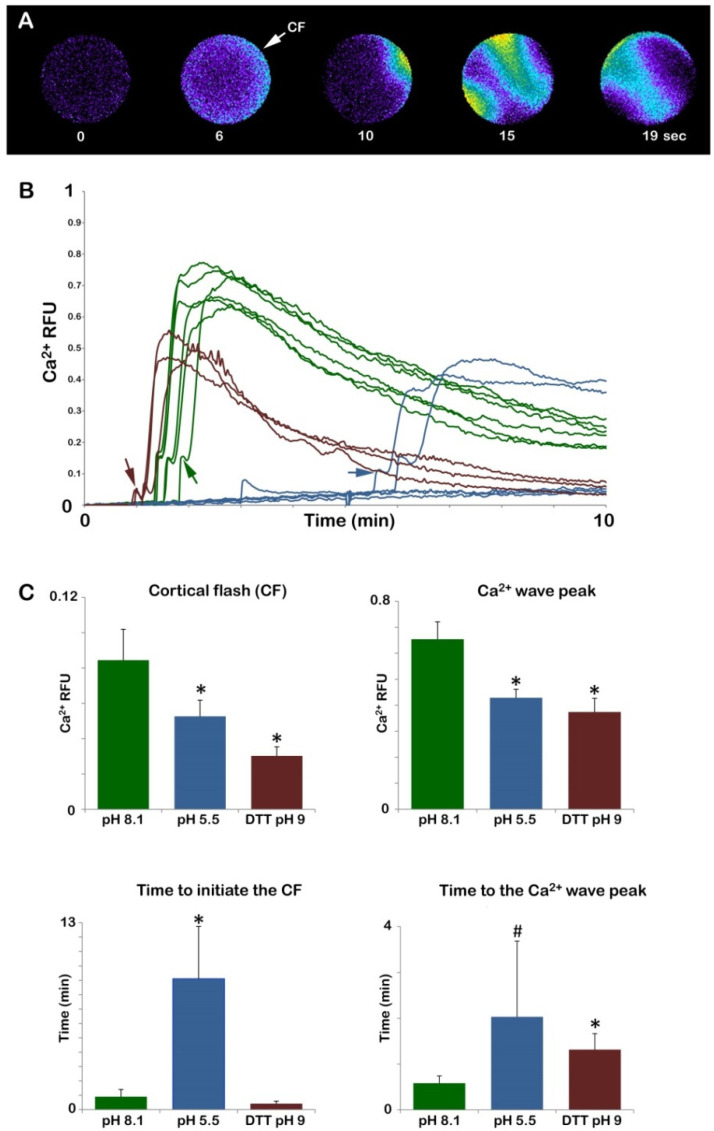
Alteration of the Ca^2+^ responses in *P. lividus* eggs fertilized after removing the jelly coat (acidic seawater) or stripping the VL (DTT, pH 9). *P. lividus* eggs were microinjected with the calcium dye before treating them with acidic seawater or 10 mM of DTT for 20 min (pH 9) to remove the vitelline layer. After several washes in NSW (pH 8.1), the Ca^2+^ response of denuded eggs was compared with that of the control eggs and the eggs inseminated in acidic seawater (pH 5.5). The pseudocolor images in (**A**) show the instantaneous increases in Ca^2+^ levels extracted from a time-lapse acquisition following fertilization of the denuded eggs, i.e., deprived of both the vitelline and jelly layers. The initial Ca^2+^ signal in the cortical region of the egg (CF) was followed by two Ca^2+^ waves as a result of polyspermic fertilization. The graphs and histograms in (B) and (C) show a comparison of the sperm-induced Ca^2+^ increases in the three different experimental conditions. The heights of the CF (arrows) and Ca^2+^ wave in the eggs fertilized in NSW (pH 8.1) (green lines) are higher than those fertilized in acidic seawater (blue lines) and the denuded eggs (red lines). The graphs and histograms (**B**,**C**) of the Ca^2+^ releases in the eggs fertilized in acidic seawater show a time delay between the addition of the sperm to the Ca^2+^ measurement chamber in which the eggs were suspended as compared to the other experimental conditions. Tukey’s post hoc test * *p* < 0.01, # *p* < 0.05. RFU = relative fluorescence unit.

**Figure 5 cells-11-02984-f005:**
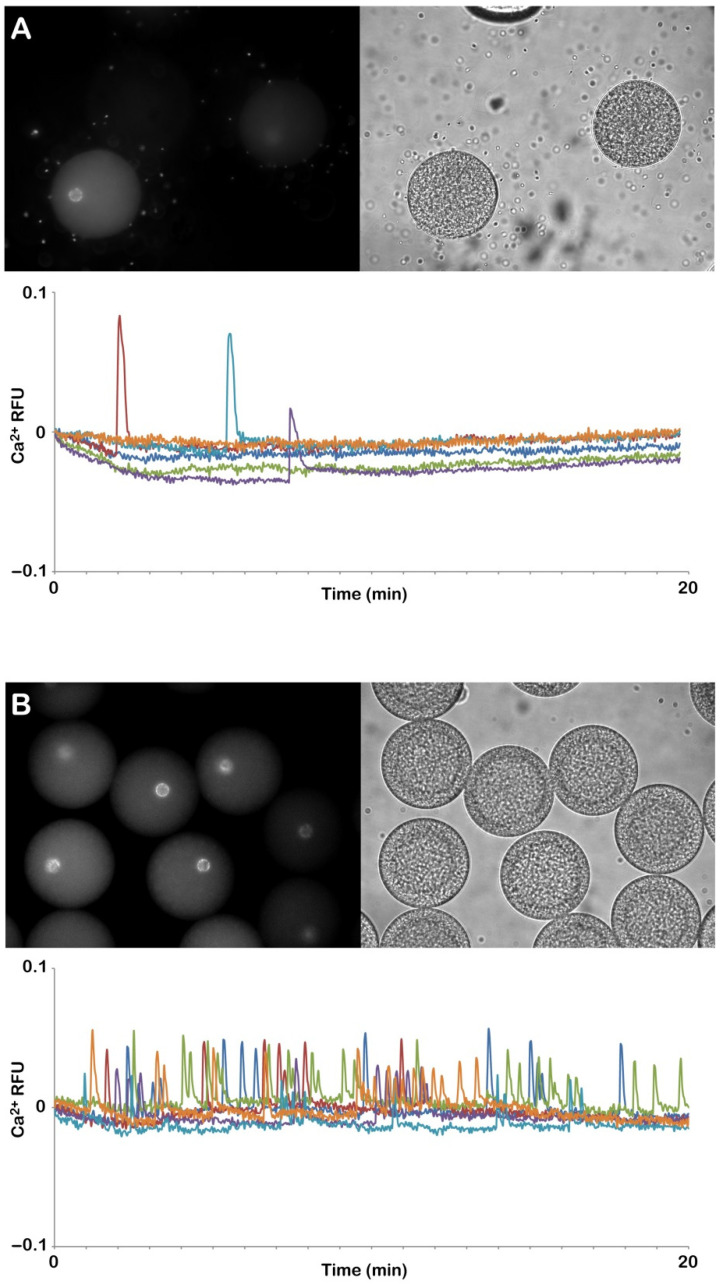
Comparison of the cross-fertilization between whole *P. lividus* and denuded eggs with *Arbacia lixula* sperm. (**A**) The fluorescent and transmitted light images show *A. lixula* sperm embedded in the jelly layer surrounding *P. lividus* eggs 5 min after insemination. The graph shows the measurement (20 min) of the Ca^2+^ changes following the interaction between gametes during heterologous fertilization. The first CF following the interaction of *A. lixula* sperm with the intact *P. lividus* eggs was detected around 5 min after the sperm addition. No Ca^2+^ signals were transduced by heterospecific sperm in 14 of 17 eggs. (**B**) Insemination of denuded *P. lividus* eggs with *A. lixula* sperm elicited transient and small Ca^2+^ changes only after the addition of sperm set at time 0, probably due to the mechanical stimulation of the sperm to the egg surface. RFU = relative fluorescence unit.

**Figure 6 cells-11-02984-f006:**
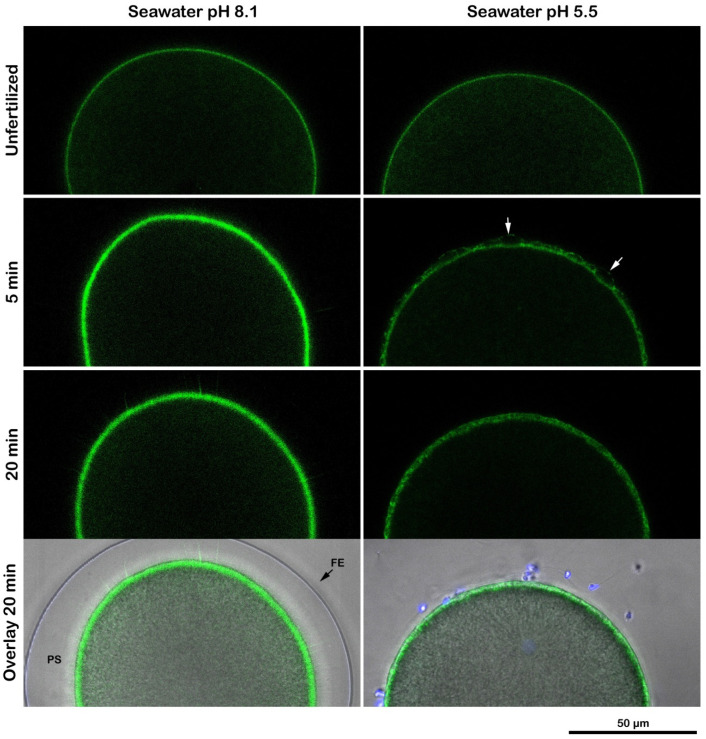
Changes of the cortical actin cytoskeleton in the jelly-free sea urchin eggs inseminated in acidic seawater (pH 5.5). *P. lividus* eggs injected with the LifeAct-GFP fusion protein were inseminated either in NSW (pH 8.1) or in acidic seawater (pH 5.5, after preincubation). In the control eggs, the cortical actin underwent polymerization to form a thick layer beneath the plasma membrane by 5 min after insemination, which persisted for 20 min. By contrast, in the eggs incubated and inseminated in acidic seawater, even if a thin layer of actin was detected in the unfertilized eggs, the lower pH of the seawater prevented the cortical actin polymerization from normally occurring 5 min after insemination. The fluorescent and overlay images show a weaker and discontinuous polymerized layer 20 min after insemination. Full elevation of the FE was compromised due to the attachment of the FE with the egg surface in several regions of the egg surface (arrows). Sperm were seen attached to the egg surface 20 min after insemination (see also Figure 2D).

**Figure 7 cells-11-02984-f007:**
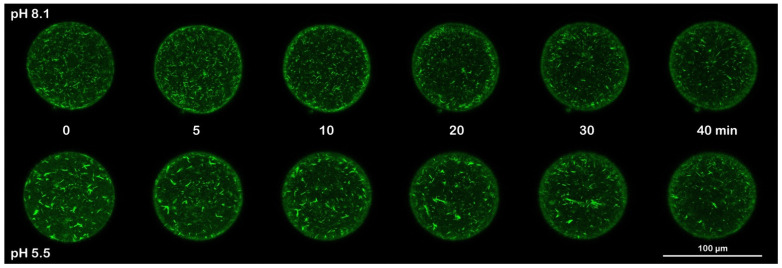
F-actin translocation following fertilization of *P. lividus* control eggs was inhibited in eggs incubated and inseminated in acidic seawater. *P. lividus* eggs were microinjected with AlexaFluor 568-Phalloidin and incubated in NSW or acidic seawater for 5 min before insemination. The addition of sperm induced F-actin changes, which were monitored by confocal microscopy at the intervals shown in the images. The moment of insemination was set as time 0. F-actin translocation was evident 20 min after the sperm addition, which coincided with the time of the actin polymerization in the egg cortex visualized by the LifeAct-GFP fusion protein shown in Figure 6.

## Data Availability

Not applicable.

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
