# Peer review of "Species-Specific Gamete Interaction during Sea Urchin Fertilization: Roles of the Egg Jelly and Vitelline Layer"

_cells, 2022, doi:10.3390/cells11192984_

Round 1

Reviewer 1 Report

The sea urchin is one of the best animals for understanding the process of fertilization because of its simple gamete structure and the vast amount of previous research. In this article the authors investigated the specific roles of the egg jelly and vitelline layer at fertilization of sea urchin gametes. They prepared dejellied and denuded (no vitelline layer) eggs and observed the ultrastructure, Ca2+ responses and Actin dynamics at insemination. Their experiments are well organized and clearly photographed. These observations will provide useful information to the readers of this journal. However, it wasn’t clear to me what points the author concluded from their observations. I would like to recommend publication of the paper after the authors have addressed the concerns which I have stated above. I hope these comments will be helpful.

The title in this article is not clear and appropriate. It would be better to change the title to reflect the conclusions drawn from the experimental results.

I didn’t understand why the authors performed the experiments under the acidic condition. The dejellied eggs incubated in acidic sea water (pH5.5) for 5 min are shrink (Fig1D) and at insemination the fertilization envelope failed to undergo a full elevation and was bound with many sperm (Fig2D). In addition, Ca2+ elevation is delayed and diminished and changes of the cortical actin polymerization also disrupted at insemination in the dejellied eggs. I wonder that these results are caused by the acidification not only due to remove the egg jelly. Sperm behavior and signaling are also affected by acidic condition. I would like to know why the authors did not perform these experiments in normal sea water at pH8.1 after the treatment of the acidic sea water for 5 min as they mentioned in L212.

The denuded (no vitelline layer) eggs prepared with DTT at pH9 for 20 min showed a disrupted microvilli morphology and multiple sperm penetration. The authors concluded that the egg plasma membrane and not the vitelline layer is the site where the specific recognition, and the vitelline layer works to prevent polyspermy in the abstract. Fig5B showed that the transient Ca2+ changes were induced by other species sperm in the denuded eggs. I didn’t understand how these results lead the conclusion. Does the Ca2+ changes in the cross-fertilization with other species sperm relate to the specific recognition or not? I think that the additional experimental data about the cross-fertilization in the denuded eggs without vitelline layer is required.

It is difficult to follow the part of L287-304. The authors referred Fig3, 2, 4 and 6 in this paragraph. I think that the authors should reorganize and rewrite the structure.

The authors reported the effects on the morphology of the microvilli in dejellied and denuded eggs before and after fertilization. However, I did not understand how microvillus morphology is related to sperm binding and polyspermy rejection. The authors should clearly state about this in results and discussion.

Minor points

Materials and methods

L130

Could you provide DTT concentrations?

Figure 4 DDT>DTT

Figure 6

No explanation about arrows.

Reviewer 2 Report

This is an interesting study that presents data to suggest that the early interactions of the key features of the way the sperm fuses and activated the sea urchin egg are not dependent on the vitelline layer.  The data is presented in generally presented in a clear manner. I have a few specific comments on the manuscripts and one aspects of the data.

11. Fig.4B shows differences in the amplitude of Ca2+ transients at fertilization depending on whether eggs were in pH 5.5 sea water or treated with DTT. The amplitudes are plotted as F-F0/F0 of Calcium Green dextran and hence they depend upon the resting Ca2+ (F0) level before fertilization.  Treating eggs with acidic sea water or DTT may alter resting Ca2+.  The authors reported injecting rhodamine along with Calcium Green, so they could plot the Ca2+ amplitudes as the Ca2+ Green over Rhodamine fluoresence, or at least use the rhodamine signal to correct for any differences in resting Ca2+ in order to be sure that the differences in Ca2+ amplitide in Fig.4 are real.

22.  The potential for sperm egg interactions at the plasma membrane to affect Ca2+ release in the egg at fertilization is one of the most interesting aspects of this study. Can the authors offer some explanation for why this effect may occur, if indeed it is a real effect?

33. The Discussion and Conclusion sections are too long. They read more like a review article in which the history of results in this area are discussed. It would be better if the authors cut down the text, used shorter paragraphs, and concentrated on the results in this paper. It would make more impact if the conclusions were a few sentences with a ‘take home message’.

Reviewer 3 Report

Limatola and colleagues present here a study regarding the interaction between sperm and eggs, and the contribution of the jelly coat and vitelline layers, using sea urchin gametes. The questions asked here are of broad interest and have been intensely investigated for many years. The authors perform some very nice electron microscopy, as well as live imaging of calcium and actin dynamics. However, the study lacks critical controls, uses harsh perturbations, is inconsistent both internally and with prior literature, and in my view does not offer a substantial advance in our understanding of fertilization. The final model that the VL and egg jelly serve as a barrier, rather than a necessary interface for sperm recognition, is confusing and doesn’t seem to fit the data here, or in the field. I therefore recommend against publication.

Major points

1.     Acid treatment has been used extensively by many labs to de-jelly eggs, for example, prior to microinjection in S. purpuratus. This has no negative effects on fertilization and development, yet the authors here report a substantial effect. I am puzzled by this incongruity. Moreover, rather than briefly treating eggs with acid seawater and then washing out, the eggs here are left in the acid seawater and fertilized therein. A prolonged treatment in this medium is harmful to the eggs and can cause numerous pleotropic effects. It could even be effecting the sperm rather than the eggs. Indeed, the authors see extensive shriveling of the eggs, indicating their manipulation has done much more than simply remove the jelly coat. Finally, it’s not clear if a wash was performed, meaning that soluble jelly could still be present and available to activate sperm.

2.     Importantly, the authors did not evaluate the extent to which jelly was removed, as they mention that jelly is not preserved in the fixation for EM. Therefore, we are left to just trust them that all the jelly has been removed. This must be directly measured, for instance by staining with ink or some other method. Otherwise, the results cannot be interpreted. There are also ways to fix the egg that maintains the jelly (Bonnell et al., 1993, for instance). Nevertheless, the authors report a much reduced fertilization rate with acid treatment, suggesting that it is important for fertilization (which we already knew).

3.     Similar caveats with DTT treatment. Prolonged treatment with a strong reducing agent and high pH will have numerous and profound effects on the egg beyond just removing the VL. Indeed, the authors mention that this damaged the eggs to the point that the actin imaging could not be performed (line 564 in the discussion). Again, it is not even clear to what extent the treatment actually removed the VL. We are shown in Fig 1 a ~2 micron wide TEM view of a single egg. But is this really representative of the whole egg population? Without some quantification scheme, it is difficult to believe. Furthermore, in the control and acid treated images, the VL indeed looks heterogeneously present. On top of this, in Fig.2 I, the supposedly VL-deficient egg has some electron density on the right-side microvilli that looks like the VL in control.

4.     The authors assess fertilization rate and polyspermy in Figure 3. From the images provided, which are wide-field fluorescence and not confocal, it is difficult to interpret whether I’m really looking at multiple sperm nuclei inside the egg, or rather an out-of-focus sperm attached to the egg surface. Also, some quantification would help here: I suggest including a histogram-style graph here that counts the instances of eggs with 0, 1, 2, 3…etc fertilizing sperm. On line 291, they write “At variance with this, only 14 out of 40 eggs incubated in seawater at pH 5.5 to remove the egg jelly and inseminated in the same medium showed monospermic fertilization”. Does this mean the others were unfertilized? Polyspermic?

5.    I appreciate the logic for the experiment in Figure 5, but I find the results to be confusing. In the bottom panel (A.l. sperm x denuded P.l. eggs), periodic calcium spiking is observed. The authors state this can be interpreted as “Ca2+ releasing events in the egg surface induced by the mechanical stimulation of the swimming sperm.” It is quite far-fetched that the minuscule forces exerted would have this mechanical effect on the egg. It may be more likely that these are autonomous calcium releases occurring after the very harsh and prolonged DTT treatment. A control calcium imaging experiment of denuded eggs alone, without sperm, should be performed to test this possibility.

6.     The LifeAct-GFP imaging in Figure 6 is not sufficiently replicated or quantified. Protein injections are quite variable from egg to egg, so it’s difficult to conclude much from this video, which the authors report is representative of only 2 experiments. I understand that live imaging is time consuming, but this should be repeated more times – at least 3 if not 5 or more eggs should be imaged and quantified, depending on the variance. Then, a pixel intensity quantification of cortical actin could be conducted and plotted at the different time points. It’s not enough to just show an image and say “representative of N eggs.”

7.     I’m confused by the phalloidin results in Figure 7. First, it is difficult to see much difference between the control and acid treated egg, and no quantification is provided. Second, I’m not sure what the authors mean by “actin fiber translocation” or “centripetal” translocation. Third, I’m very skeptical of using phalloidin in a live-cell assay, because it strongly perturbs actin dynamics by stabilizing filaments, despite the prior literature cited. Given its perturbative properties, and the fact that we don’t see these huge actin bundles in the previously LifeAct experiment, I question whether they are physiological or instead an artefact of phalloidin injection.

Minor points

1.     The writing is extremely dense and full of long run-on sentences that are difficult to follow. As a result, the authors’ arguments often get lost. There are many confusing passages throughout, but one example: we don’t reach the point of the paper in the in the discussion until the very end (line 572) “Thus at variance with the data in the literature, the results from this work suggest that the intact vitelline layer covering the egg plasma membrane of unfertilized eggs acts as a barrier…” I suggest bringing statements like this to the foreground, and working to eliminate unnecessary information.

2.     The title is very vague and doesn’t really tell me what the paper is about.

3.     The abstract has several sentences regarding prior work in the field that are presented in a way to make them sound like they are findings of this study (“We learned that…alkaline seawater”). This could probably be eliminated entirely.

4.     Please present the TEM images in Figure 2 at the same zoom level so they can be more easily compared.

5.     In Fig. 4A, please show images for also control and acid treated eggs. Otherwise, this figure is nicely quantified and presented.

6.     Consider including a model slide to help the reader understand concepts from this paper and prior work.

Round 2

Reviewer 3 Report

The authors have submitted very light textual edits. However, they have chosen not to include any new controls, perform any quantification, or modify the figure presentation to address any of my concerns with the original manuscript. I found the justification for these omissions in the reviewer response to be confusing and unconvincing. Unfortunately, I am unable to change my original recommendation of rejection for this study. Rather than reiterate all of my original comments, I list my main takeaways below:

1.     I’m more confused than before about how the acid treatment was performed. What does washed “as needed” mean? In their response, the authors argue that since acid treatment followed by release into normal sea water (the standard practice for removing jelly) is indistinguishable from control, they decided not to show the results. However, this would appear to support my original concern, which is that the fertilization phenotype is not due to removing egg jelly per se, but rather an unintended consequence of fertilizing the eggs at an unnaturally low pH.   

2.     Critical controls have been left out, which makes it impossible to know if the manipulations are doing what the authors claim. For instance, they argue against including experiments to evaluate whether jelly has been fully removed (an essential criteria for interpreting the results). Instead, they mention that the “egg-to-egg distance was nearly reduced to none”, but do not show us the data. They also offer no quantification for vitelline layer removal. 

In another example, they do not include an unfertilized control for Figure 5B, and only show data after fertilization. To know whether the spiking behavior is due to sperm addition, we need to see the situation without sperm for an equivalent length of time. Not including an image of a normal calcium wave (Fig 4A) because they showed it “numerous times in our previous papers” with different batches of eggs is not an appropriate way to conduct a control.

3.     Despite their insistence, it is difficult to imagine from a biochemical perspective the idea that 10mM DTT is not having other effects beyond removing the vitelline layer. For instance, DTT at this concentration stimulates cell cycle progression in various echinoderm eggs. Similarly, injection of phalloidin will have unintended consequences. The references cited to refute this are subject to the same caveats, in my view.

4.     There are numerous data representation issues. I have no doubt that the authors repeated their experiments, but these results must be quantified, analyzed, and presented in the figures, not just briefly mentioned in the text that they repeated it. For example, no microinjection experiment is so precise, and no biological variation is so low, that this can be ignored. It is basic acceptable practice in cell biology. As presented, it is not possible to discern any meaningful effect on the actin networks in figures 6 and 7.

5.     Given all the above caveats, the substantial claim that the plasma membrane is the site of sperm-egg recognition needs a much higher level of support than is currently offered. Moreover, the mechanistic links are between acid treatment and the downstream effects on the actin cytoskeleton are quite tenuous.